# A Tractable Approximation to Optimal Point Process Filtering: Application to Neural Encoding

**Yuval Harel, Ron Meir**
Department of Electrical Engineering
Technion – Israel Institute of Technology
Technion City, Haifa, Israel
{yharel@tx,rmeir@ee}.technion.ac.il

**Manfred Opper**
Department of Artificial Intelligence
Technical University Berlin
Berlin 10587, Germany
opperm@cs.tu-berlin.de

## Abstract

The process of dynamic state estimation (filtering) based on point process observations is in general intractable. Numerical sampling techniques are often practically useful, but lead to limited conceptual insight about optimal encoding/decoding strategies, which are of significant relevance to Computational Neuroscience. We develop an analytically tractable Bayesian approximation to optimal filtering based on point process observations, which allows us to introduce distributional assumptions about sensory cell properties, that greatly facilitate the analysis of optimal encoding in situations deviating from common assumptions of uniform coding. The analytic framework leads to insights which are difficult to obtain from numerical algorithms, and is consistent with experiments about the distribution of tuning curve centers. Interestingly, we find that the information gained from the absence of spikes may be crucial to performance.

## 1   Introduction

The task of inferring a hidden dynamic state based on partial noisy observations plays an important role within both applied and natural domains. A widely studied problem is that of online inference of the hidden state at a given time based on observations up to to that time, referred to as *filtering* [1]. For the linear setting with Gaussian noise and quadratic cost, the solution is well known since the early 1960s both for discrete and continuous times, leading to the celebrated Kalman and the Kalman-Bucy filters [2, 3], respectively. In these cases the exact posterior distribution is Gaussian, resulting in closed form recursive update equations for the mean and variance of this distribution, implying finite-dimensional filters. However, beyond some very specific settings [4], the optimal filter is infinite-dimensional and impossible to compute in closed form, requiring either approximate analytic techniques (e.g., the extended Kalman filter (e.g., [1]), the unscented filter [5]) or numerical procedures (e.g., particle filters [6]). The latter usually require time discretization and a finite number of particles, resulting in loss of precision . For many practical tasks (e.g., queuing [7] and optical communication [8]) and biologically motivated problems (e.g., [9]) a natural observation process is given by a point process observer, leading to a nonlinear infinite-dimensional optimal filter (except in specific settings, e.g., finite state spaces, [7, 10]).

We consider a continuous-state and continuous-time multivariate hidden Markov process observed through a set of sensory neuron-like elements characterized by multi-dimensional unimodal tuning functions, representing the elements' average firing rate. The tuning function parameters are characterized by a distribution allowing much flexibility. The actual firing of each cell is random and is given by a Poisson process with rate determined by the input and by the cell's tuning function. Inferring the hidden state under such circumstances has been widely studied within the Computational Neuroscience literature, mostly for static stimuli. In the more challenging and practically important dynamic setting, much work has been devoted to the development of numerical sampling techniques

for fast and effective approximation of the posterior distribution (e.g., [11]). In this work we are less concerned with algorithmic issues, and more with establishing closed-form analytic expressions for an approximately optimal filter (see [10, 12, 13] for previous work in related, but more restrictive settings), and using these to characterize the nature of near-optimal encoders, namely determining the structure of the tuning functions for optimal state inference. A significant advantage of the closed form expressions over purely numerical techniques is the insight and intuition that is gained from them about *qualitative* aspects of the system. Moreover, the leverage gained by the analytic computation contributes to reducing the variance inherent to Monte Carlo approaches. Technically, given the intractable infinite-dimensional nature of the posterior distribution, we use a projection method replacing the full posterior at each point in time by a projection onto a simple family of distributions (Gaussian in our case). This approach, originally developed in the Filtering literature [14, 15], and termed Assumed Density Filtering (ADF), has been successfully used more recently in Machine Learning [16, 17]. As far as we are aware, this is the first application of this methodology to point process filtering.

The main contributions of the paper are the following: (i) Derivation of closed form recursive expressions for the continuous time posterior mean and variance within the ADF approximation, allowing for the incorporation of distributional assumptions over sensory variables. (ii) Characterization of the optimal tuning curves (encoders) for sensory cells in a more general setting than hitherto considered. Specifically, we study the optimal shift of tuning curve centers, providing an explanation for observed experimental phenomena [18]. (iii) Demonstration that absence of spikes is informative, and that, depending on the relationship between the tuning curve distribution and the dynamic process (the 'prior'), may significantly improve the inference. This issue has not been emphasized in previous studies focusing on homogeneous populations.

We note that most previous work in the field of neural encoding/decoding has dealt with static observations and was based on the Fisher information, which often leads to misleading qualitative results (e.g., [19, 20]). Our results address the full dynamic setting in continuous time, and provide results for the posterior variance, which is shown to yield an excellent approximation of the posterior Mean Square Error (MSE). Previous work addressing non-uniform distributions over tuning curve parameters [21] used static univariate observations and was based on Fisher information rather than the MSE itself.

## 2 Problem formulation

### 2.1 Dense Gaussian neural code

We consider a dynamical system with state $X_t \in \mathbb{R}^n$, observed through an observation process $N$ describing the firing patterns of sensory neurons in response to the process $X$. The observed process is a diffusion process obeying the Stochastic Differential Equation (SDE)

$$dX_t = A\left(X_t\right)dt + D\left(X_t\right)dW_t, \quad (t \geq 0)$$

where $A\left(\cdot\right), D\left(\cdot\right)$ are arbitrary functions and $W_t$ is standard Brownian motion. The initial condition $X_0$ is assumed to have a continuous distribution with a known density. The observation process $N$ is a marked point process [8] defined on $[0, \infty) \times \mathbb{R}^m$, meaning that each point, representing the firing of a neuron, is identified by its time $t \in [0, \infty)$, and a mark $\theta \in \mathbb{R}^m$. In this work the mark is interpreted as a parameter of the firing neuron, which we refer to as the neuron's *preferred stimulus*. Specifically, a neuron with parameter $\theta$ is taken to have firing rate

$$\lambda\left(x; \theta\right) = h \exp\left(-\frac{1}{2}\left\|Hx - \theta\right\|_{\Sigma_{\text{tc}}^{-1}}^2\right),$$

in response to state $x$, where $H \in \mathbb{R}^{m \times n}$ and $\Sigma_{\text{tc}} \in \mathbb{R}^{m \times m}$, $m \leq n$, are fixed matrices, and the notation $\left\|y\right\|_M^2$ denotes $y^T M y$. The choice of Gaussian form for $\lambda$ facilitates analytic tractability. The inclusion of the matrix $H$ allows using high-dimensional models where only some dimensions are observed, for example when the full state includes velocities but only locations are directly observable. We also define $N_t \triangleq N\left([0, t) \times \mathbb{R}^m\right)$, i.e., $N_t$ is the total number of points up to time $t$, regardless of their location $\theta$, and denote by $\mathcal{N}_t$ the *sequence* of points up to time $t$ — formally,

the process $N$ restricted to $[0, t) \times \mathbb{R}^m$. Following [8], we use the notation

$$\int_a^b \int_U f(t, \theta) N(dt \times d\theta) \triangleq \sum_i \mathbf{1} \{t_i \in [a, b], \theta_i \in U\} f(t_i, \theta_i),$$  (1)

for $U \subseteq \mathbb{R}^m$ and any function $f$, where $(t_i, \theta_i)$ are respectively the time and mark of the $i$-th point of the process $N$.

Consider a network with $M$ sensory neurons, having random preferred stimuli $\boldsymbol{\theta} = \{\theta_i\}_{i=1}^M$ that are drawn independently from a common distribution with probability density $f(\theta)$, which we refer to as the *population density*. Positing a distribution for the preferred stimuli allows us to obtain simple closed form solutions, and to optimize over distribution parameters rather than over the higher-dimensional space of all the $\theta_i$. The total rate of spikes with preferred stimuli in a set $A \subset \mathbb{R}^m$, given $X_t = x$, is then $\lambda_A(x; \boldsymbol{\theta}) = h \sum_i 1_{\{\theta_i \in A\}} \exp\left(-\frac{1}{2} \|Hx - \theta_i\|_{\Sigma_{tc}^{-1}}^2\right)$. Averaging over $f(\theta)$, we have the expected rate $\lambda_A(x) \triangleq \mathrm{E}\lambda_A(x; \boldsymbol{\theta}) = hM \int_A f(\theta) \exp\left(-\frac{1}{2} \|Hx - \theta\|_{\Sigma_{tc}^{-1}}^2\right) d\theta$. We now obtain an infinite neural network by considering the limit $M \to \infty$ while holding $\lambda^0 = hM$ fixed. In the limit we have $\lambda_A(x; \boldsymbol{\theta}) \to \lambda_A(x)$, so that the process $N$ has density

$$\lambda_t(\theta, X_t) = \lambda^0 f(\theta) \exp\left(-\frac{1}{2} \|HX_t - \theta\|_{\Sigma_{tc}^{-1}}^2\right),$$  (2)

meaning that the expected number of points in a small rectangle $[t, t + dt] \times \prod_i [\theta_i, \theta_i + d\theta_i]$, conditioned on the history $X_{[0,t]}, \mathcal{N}_t$, is $\lambda_t(\theta, X_t) dt \prod_i d\theta_i + o(dt, |d\theta|)$. A finite network can be obtained as a special case by taking $f$ to be a sum of delta functions.

For analytic tractability, we assume that $f(\theta)$ is Gaussian with center $c$ and covariance $\Sigma_{\text{pop}}$, namely $f(\theta) = \mathcal{N}(\theta; c, \Sigma_{\text{pop}})$. We refer to $c$ as the *population center*. Previous work [22, 20, 23] considered the case where neurons' preferred stimuli uniformly cover the space, obtained by removing the factor $f(\theta)$ from (2). Then, the total firing rate $\int \lambda_t(\theta, x) d\theta$ is independent of $x$, which simplifies the analysis, and leads to a Gaussian posterior (see [22]). We refer to the assumption that $\int \lambda_t(\theta, x) d\theta$ is independent of $x$ as *uniform coding*. The uniform coding case may be obtained from our model by taking the limit $\Sigma_{\text{pop}}^{-1} \to 0$ with $\lambda^0/\sqrt{\det \Sigma_{\text{pop}}}$ held constant.

## 2.2   Optimal encoding and decoding

We consider the question of optimal encoding and decoding under the above model. The process of neural decoding is assumed to compute (exactly or approximately) the full posterior distribution of $X_t$ given $\mathcal{N}_t$. The problem of neural encoding is then to choose the parameters $\phi = (c, \Sigma_{\text{pop}}, \Sigma_{\text{tc}})$, which govern the statistics of the observation process $N$, given a specific decoding scheme.

To quantify the performance of the encoding-decoding system, we summarize the result of decoding using a single estimator $\hat{X}_t = \hat{X}_t(\mathcal{N}_t)$, and define the Mean Square Error (MSE) as $\epsilon_t \triangleq \text{trace}[(X_t - \hat{X}_t)(X_t - \hat{X}_t)^T]$. We seek $\hat{X}_t$ and $\phi$ that solve $\min_\phi \lim_{t \to \infty} \min_{\hat{X}_t} \mathrm{E}[\epsilon_t] = \min_\phi \lim_{t \to \infty} \mathrm{E}[\min_{\hat{X}_t} \mathrm{E}[\epsilon_t | \mathcal{N}_t]]$. The inner minimization problem in this equation is solved by the MSE-optimal decoder, which is the posterior mean $\hat{X}_t = \mu_t \triangleq \mathrm{E}[X_t | \mathcal{N}_t]$. The posterior mean may be computed from the full posterior obtained by decoding. The outer minimization problem is solved by the optimal encoder. In principle, the encoding/decoding problem can be solved for any value of $t$. In order to assess performance it is convenient to consider the steady-state limit $t \to \infty$ for the encoding problem.

Below, we find a closed form approximate solution to the decoding problem for any $t$ using ADF. We then explore the problem of choosing the steady-state optimal encoding parameters $\phi$ using Monte Carlo simulations. Note that if decoding is exact, the problem of optimal encoding becomes that of minimizing the expected posterior variance.

## 3 Neural decoding

### 3.1 Exact filtering equations

Let $P\left(\cdot, t\right)$ denote the posterior density of $X_t$ given $\mathcal{N}_t$, and $\mathrm{E}_P^t\left[\cdot\right]$ the posterior expectation given $\mathcal{N}_t$. The prior density $P\left(\cdot, 0\right)$ is assumed to be known.

The problem of filtering a diffusion process $X$ from a doubly stochastic Poisson process driven by $X$ is formally solved in [24]. The result is extended to marked point processes in [22], where the authors derive a stochastic PDE for the posterior density[1],

$$dP\left(x, t\right) = \mathcal{L}^* P\left(x, t\right) dt + P\left(x, t\right) \int_{\theta \in \mathbb{R}^m} \frac{\lambda_t\left(\theta, x\right) - \hat{\lambda}_t\left(\theta\right)}{\hat{\lambda}_t\left(\theta\right)} \left(N\left(dt \times d\theta\right) - \hat{\lambda}_t\left(\theta\right) d\theta\, dt\right), \quad (3)$$

where the integral with respect to $N$ is interpreted as in (1), $\mathcal{L}$ is the state's infinitesimal generator (Kolmogorov's backward operator), defined as $\mathcal{L}f\left(x\right) = \lim_{\Delta t \to 0^+} \left(\mathrm{E}\left[f\left(X_{t+\Delta t}\right)|X_t = x\right] - f\left(x\right)\right)/\Delta t$, $\mathcal{L}^*$ is $\mathcal{L}$'s adjoint operator (Kolmogorov's forward operator), and $\hat{\lambda}_t\left(\theta\right) \triangleq \mathrm{E}_P^t\left[\lambda_t\left(\theta, X_t\right)\right] = \int P\left(x, t\right) \lambda_t\left(\theta, x\right) dx$.

The stochastic PDE (3) is usually intractable. In [22, 23] the authors consider linear dynamics with uniform coding and Gaussian prior. In this case, the posterior is Gaussian, and (3) leads to closed form ODEs for its moments. When the uniform coding assumption is violated, the posterior is no longer Gaussian. Still, we can obtain exact equations for the posterior moments, as follows.

Let $\mu_t = \mathrm{E}_P^t X_t, \tilde{X}_t = X_t - \mu_t, \Sigma_t = \mathrm{E}_P^t[\tilde{X}_t \tilde{X}_t^T]$. Using (3), and the known results for $\mathcal{L}$ for diffusion processes (see supplementary material), the first two posterior moments can be shown to obey the following equations between spikes (see [23] for the finite population case):

$$\begin{aligned}
\frac{d\mu_t}{dt} &= \mathrm{E}_P^t\left[A\left(X_t\right)\right] + \mathrm{E}_P^t\left[X_t \int \left(\hat{\lambda}_t\left(\theta\right) - \lambda_t\left(\theta, X_t\right)\right) d\theta\right] \\
\frac{d\Sigma_t}{dt} &= \mathrm{E}_P^t\left[A\left(X_t\right)\tilde{X}_t^T\right] + \mathrm{E}_P^t\left[\tilde{X}_t A\left(X_t\right)^T\right] + \mathrm{E}_P^t\left[D\left(X_t\right) D\left(X_t\right)^T\right] \\
&\quad + \mathrm{E}_P^t\left[\tilde{X}_t \tilde{X}_t^T \int \left(\hat{\lambda}_t\left(\theta\right) - \lambda_t\left(\theta, X_t\right)\right) d\theta\right].
\end{aligned} \quad (4)$$

### 3.2 ADF approximation

While equations (4) are exact, they are not practical, since they require computation of $\mathrm{E}_P^t\left[\cdot\right]$. We now proceed to find an approximate closed form for (4). Here we present the main ideas of the derivation. The formulation presented here assumes, for simplicity, an open-loop setting where the system is passively observed. It can be readily extended to a closed-loop control-based setting, and is presented in this more general framework in the supplementary material, including full details.

To bring (4) to a closed form, we use ADF with an assumed Gaussian density (see [16] for details). Conceptually, this may be envisioned as integrating (4) while replacing the distribution $P$ by its approximating Gaussian "at each time step". Assuming the moments are known exactly, the Gaussian is obtained by matching the first two moments of $P$ [16]. Note that the solution of the resulting equations does not in general match the first two moments of the exact solution, though it may approximate it.

Abusing notation, in the sequel we use $\mu_t, \Sigma_t$ to refer to the ADF approximation rather than to the exact values. Substituting the normal distribution $\mathcal{N}(x; \mu_t, \Sigma_t)$ for $P(x, t)$ to compute the expectations involving $\lambda_t$ in (4), and using (2) and the Gaussian form of $f(\theta)$, results in computable Gaussian integrals. Other terms may also be computed in closed form if the function $A, D$ can be expanded as power series. This computation yields approximate equations for $\mu_t, \Sigma_t$ between spikes. The updates at spike times can similarly be computed in closed form either from (3) or directly from a Bayesian update of the posterior (see supplementary material, or e.g., [13]).

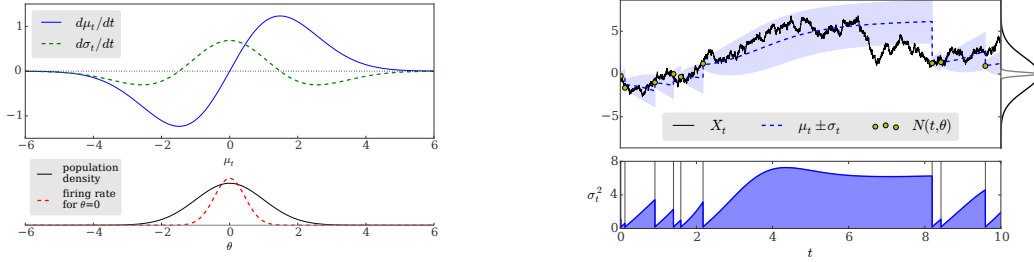

Figure 1: **Left** Changes to the posterior moments between spikes as a function of the current posterior mean estimate, for a static 1-d state. The parameters are $a = d = 0, H = 1, \sigma^2_{\text{pop}} = 1, \sigma^2_{\text{tc}} = 0.2, c = 0, \lambda^0 = 10, \sigma_t = 1$. The bottom plot shows the density of preferred stimuli $f(\theta)$ and tuning curve for a neuron with preferred stimulus $\theta = 0$. **Right** An example of filtering a linear one-dimensional process. Each dot correspond to a spike with the vertical location indicating the preferred stimulus $\theta$. The curves to the right of the graph show the preferred stimulus density (black), and a tuning curve centered at $\theta = 0$ (gray). The tuning curve and preferred stimulus density are normalized to the same height for visualization. The bottom graph shows the posterior variance, with the vertical lines showing spike times. Parameters are: $a = -0.1, d = 2, H = 1, \sigma^2_{\text{pop}} = 2, \sigma^2_{\text{tc}} = 0.2, c = 0, \lambda^0 = 10, \mu_0 = 0, \sigma^2_0 = 1$. Note the decrease of the posterior variance following $t = 4$ even though no spikes are observed.

For simplicity, we assume that the dynamics are linear, $dX_t = AX_t\, dt + D\, dW_t$, resulting in the filtering equations

$$d\mu_t = A\mu_t dt + g_t \Sigma_t H^T S_t (H\mu_t - c)\, dt + \Sigma_{t^-} H^T S^{\text{tc}}_{t^-} \int_{\theta \in \mathbb{R}^m} (\theta - H\mu_{t^-})\, N(dt \times d\theta) \quad (5)$$

$$
\begin{aligned}
d\Sigma_t = {} & \left(A\Sigma_t + \Sigma_t A^T + DD^T\right) dt \\
& + g_t \Sigma_t H^T \left[S_t - S_t (H\mu_t - c)(H\mu_t - c)^T S_t\right] H\Sigma_t dt \\
& - \Sigma_{t^-} H^T S^{\text{tc}}_{t^-} H\Sigma_{t^-} dN_t,
\end{aligned}
\quad (6)
$$

where $S^{\text{tc}}_t \triangleq \left(\Sigma_{\text{tc}} + H\Sigma_t H^T\right)^{-1}$, $S_t \triangleq \left(\Sigma_{\text{tc}} + \Sigma_{\text{pop}} + H\Sigma_t H^T\right)^{-1}$, and

$$g_t \triangleq \int \hat{\lambda}(\theta)\, d\theta = \int \mathrm{E}^t_P[\lambda(\theta, X_t)]\, d\theta = \lambda^0 \sqrt{\det(\Sigma_{\text{tc}} S_t)} \exp\left(-\frac{1}{2}\|H\mu_t - c\|^2_{S_t}\right)$$

is the posterior expected total firing rate. Expressions including $t^-$ are to be interpreted as left limits $f(t^-) = \lim_{s \to t^-} f(s)$, which are necessary since the solution is discontinuous at spike times.

The last term in (5) is to be interpreted as in (1). It contributes an instantaneous jump in $\mu_t$ at the time of a spike with preferred stimulus $\theta$, moving $H\mu_t$ closer to $\theta$. Similarly, the last term in (6) contributes an instantaneous jump in $\Sigma_t$ at each spike time, which is the same regardless of spike location. All other terms describe the evolution of the posterior between spikes: the first few terms in (5)-(6) are the same as in the dynamics of the prior, as in [13, 23], whereas the terms involving $g_t$ correspond to information from the absence of spikes. Note that the latter scale with $g_t$, the expected total firing rate, i.e., lack of spikes becomes "more informative" the higher the expected rate of spikes.

It is illustrative to consider these equations in the scalar case $m = n = 1$, with $H = 1$. Letting $\sigma^2_t = \Sigma_t, \sigma^2_{\text{tc}} = \Sigma_{\text{tc}}, \sigma^2_{\text{pop}} = \Sigma_{\text{pop}}, a = A, d = D$ yields

$$d\mu_t = a\mu_t dt + g_t \frac{\sigma^2_t}{\sigma^2_t + \sigma^2_{\text{tc}} + \sigma^2_{\text{pop}}}(\mu_t - c)\, dt + \frac{\sigma^2_{t^-}}{\sigma^2_{t^-} + \sigma^2_{\text{tc}}} \int_{\theta \in \mathbb{R}} (\theta - \mu_{t^-})\, N(dt \times d\theta) \quad (7)$$

$$d\sigma^2_t = \left(2a\sigma^2_t + d^2 + g_t \frac{\sigma^2_t}{\sigma^2_t + \sigma^2_{\text{tc}} + \sigma^2_{\text{pop}}}\left[1 - \frac{(\mu_t - c)^2}{\sigma^2_t + \sigma^2_{\text{tc}} + \sigma^2_{\text{pop}}}\right]\sigma^2_t\right) dt - \frac{\sigma^2_{t^-}}{\sigma^2_{t^-} + \sigma^2_{\text{tc}}}\sigma^2_{t^-}\, dN_t, \quad (8)$$

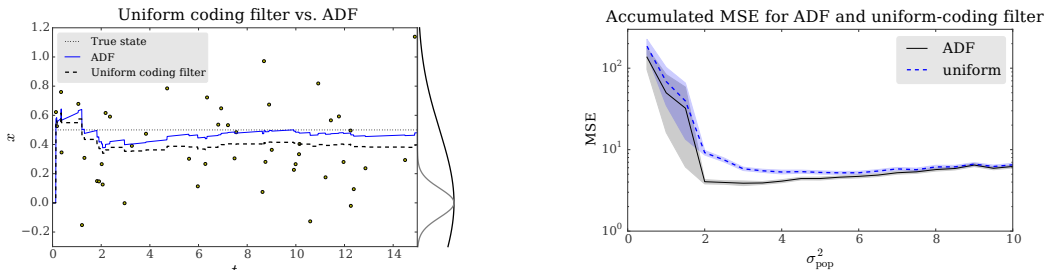

Figure 2: **Left** Illustration of information gained between spikes. A static state $X_t = 0.5$, shown in a dotted line, is observed and filtered twice: with the correct value $\sigma^2_{\text{pop}} = 0.5$ ("ADF", solid blue line), and with $\sigma^2_{\text{pop}} = \infty$ ("Uniform coding filter", dashed line). The curves to the right of the graph show the preferred stimulus density (black), and a tuning curve centered at $\theta = 0$ (gray). Both filters are initialized with $\mu_0 = 0, \sigma^2_0 = 1$. **Right** Comparison of MSE for the ADF filter and the uniform coding filter. The vertical axis shows the integral of the square error integrated over the time interval $[5, 10]$, averaged over 1000 trials. Shaded areas indicate estimated errors, computed as the sample standard deviation divided by the square root of the number of trials. Parameters in both plots are $a = d = 0, c = 0, \sigma^2_{\text{pop}} = 0.5, \sigma^2_{\text{tc}} = 0.1, H = 1, \lambda^0 = 10$.

where $g_t = \lambda^0 \sqrt{2\pi\sigma^2_{\text{tc}}} \mathcal{N}\left(\mu_t; c, \sigma^2_t + \sigma^2_{\text{tc}} + \sigma^2_{\text{pop}}\right)$. Figure 1 (left) shows how $\mu_t, \sigma^2_t$ change between spikes for a static 1-dimensional state ($a = d = 0$). In this case, all terms in the filtering equations drop out except those involving $g_t$. The term involving $g_t$ in $d\mu_t$ pushes $\mu_t$ away from $c$ in the absence of spikes. This effect weakens as $|\mu_t - c|$ grows due to the factor $g_t$, consistent with the idea that far from $c$, the lack of spikes is less surprising, hence less informative. The term involving $g_t$ in $d\sigma^2_t$ increases the variance when $\mu_t$ is near $c$, otherwise decreases it.

### 3.3 Information from lack of spikes

An interesting aspect of the filtering equations (5)-(6) is that the dynamics of the posterior density between spikes differ from the prior dynamics. This is in contrast to previous models which assumed uniform coding: the (exact) filtering equations appearing in [22] and [23] have the same form as (5)-(6) except that they do not include the correction terms involving $g_t$, so that between spikes the dynamics are identical to the prior dynamics. This reflects the fact that lack of spikes in a time interval is an indication that the total firing rate is low; in the uniform coding case, this is not informative, since the total firing rate is independent of the state.

Figure 2 (left) illustrates the information gained from lack of spikes. A static scalar state is observed by a process with rate (2), and filtered twice: once with the correct value of $\sigma_{\text{pop}}$, and once with $\sigma_{\text{pop}} \to \infty$, as in the uniform coding filter of [23]. Between spikes, the ADF estimate moves away from the population center $c = 0$, whereas the uniform coding estimate remains fixed. The size of this effect decreases with time, as the posterior variance estimate (not shown) decreases. The reduction in filtering errors gained from the additional terms in (5)-(6) is illustrated in Figure 2 (right). Despite the approximation involved, the full filter significantly outperforms the uniform coding filter. The difference disappears as $\sigma_{\text{pop}}$ increases and the population becomes uniform.

**Special cases** To gain additional insight into the filtering equations, we consider their behavior in several limits. (i) As $\sigma^2_{\text{pop}} \to \infty$, spikes become rare as the density $f(\theta)$ approaches 0 for any $\theta$. The total expected rate of spikes $g_t$ also approaches 0, and the terms corresponding to information from lack of spikes vanish. Other terms in the equations are unaffected. (ii) In the limit $\sigma^2_{\text{tc}} \to \infty$, each neuron fires as a Poisson process with a constant rate independent of the observed state. The total expected firing rate $g_t$ saturates at its maximum, $\lambda^0$. Therefore the preferred stimuli of spiking neurons provide no information, nor does the presence or absence of spikes. Accordingly, all terms other than those related to the prior dynamics vanish. (iii) The uniform coding case [22, 23] is obtained as a special case in the limit $\sigma^2_{\text{pop}} \to \infty$ with $\lambda^0/\sigma_{\text{pop}}$ constant. In this limit the terms involving $g_t$ drop out, recovering the (exact) filtering equations in [22].

# 4 Optimal neural encoding

We model the problem of optimal neural encoding as choosing the parameters $c, \Sigma_{\text{pop}}, \Sigma_{\text{tc}}$ of the population and tuning curves, so as to minimize the steady-state MSE. As noted above, when the estimate is exactly the posterior mean, this is equivalent to minimizing the steady-state expected posterior variance. The posterior variance has the advantage of being less noisy than the square error itself, since by definition it is the mean of the square error (of the posterior mean) under conditioning by $\mathcal{N}_t$. We explore the question of optimal neural encoding by measuring the steady-state variance through Monte Carlo simulations of the system dynamics and the filtering equations (5)-(6). Since the posterior mean and variance computed by ADF are approximate, we verified numerically that the variance closely matches the MSE in the steady state when averaged across many trials (see supplementary material), suggesting that asymptotically the error in estimating $\mu_t$ and $\Sigma_t$ is small.

## 4.1 Optimal population center

We now consider the question of the optimal value for the population center $c$. Intuitively, if the prior distribution of the process $X$ is unimodal with mode $x_0$, the optimal population center is at $Hx_0$, to produce the most spikes. On the other hand, the terms involving $g_t$ in the filtering equation (5)-(6) suggest that the lack of spikes is also informative. Moreover, as seen in Figure 1 (left), the posterior variance is reduced between spikes only when the current estimate is far enough from $c$. These considerations suggest that there is a trade-off between maximizing the frequency of spikes and maximizing the information obtained from lack of spikes, yielding an optimal value for $c$ that differs from $Hx_0$.

We simulated a simple one-dimensional process to determine the optimal value of $c$ which minimizes the approximate posterior variance $\Sigma_t$. Figure 3 (left) shows the posterior variance for varying values of the population center $c$ and base firing rate $\lambda^0$. For each firing rate, we note the value of $c$ minimizing the posterior variance (the optimal population center), as well as the value of $c_{\text{m}} = \operatorname{argmin}_c (d\sigma_t/dt|_{\mu_t=0})$, which maximizes the reduction in the posterior variance when the current state estimate $\mu_t$ is at the process equilibrium $x_0 = 0$. Consistent with the discussion above, the optimal value lies between 0 (where spikes are most abundant) and $c_{\text{m}}$ (where lack of spikes is most informative). As could be expected, the optimal center is closer to 0 the higher the base firing rate. Similarly, wide tuning curves, which render the spikes less informative, lead to an optimal center farther from 0 (Figure 3, right).

A shift of the population center relative to the prior mode has been observed physiologically in encoding of inter-aural time differences for localization of sound sources [25]. In [18], this phenomenon was explained in a finite population model based on maximization of Fisher information. This is in contrast to the results of [21], which consider a heterogeneous population where the tuning curve width scales roughly inversely with neuron density. In this case, the population density maximizing the Fisher information is shown to be monotonic with the prior, i.e., more neurons should be assigned to more probable states. This apparent discrepancy may be due to the scaling of tuning curve widths in [21], which produces roughly constant total firing rate, i.e., uniform coding. This demonstrates that a non-constant total firing rate, which renders lack of spikes informative, may be necessary to explain the physiologically observed shift phenomenon.

## 4.2 Optimization of population distribution

Next, we consider the optimization of the population distribution, namely, the simultaneous optimization of the population center $c$ and the population variance $\Sigma_{\text{pop}}$ in the case of a static scalar state. Previous work using a finite neuron population and a Fisher information-based criterion [18] has shown that the optimal distribution of preferred stimuli depends on the prior variance. When it is small relative to the tuning curve width, optimal encoding is achieved by placing all preferred stimuli at a fixed distance from the prior mean. On the other hand, when the prior variance is large relative to the tuning curve width, optimal encoding is uniform (see figure 2 in [18]).

Similar results are obtained with our model, as shown in Figure 4. Here, a static scalar state drawn from $\mathcal{N}(0, \sigma_{\text{p}}^2)$ is filtered by a population with tuning curve width $\sigma_{\text{tc}} = 1$ and preferred stimulus density $\mathcal{N}(c, \sigma_{\text{pop}}^2)$. In Figure 4 (left), the prior distribution is narrow relative to the tuning curve width, leading to an optimal population with a narrow population distribution far from the origin. In

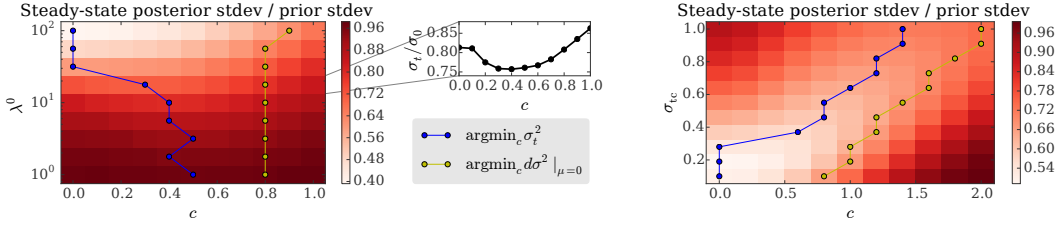

Figure 3: Optimal population center location for filtering a linear one-dimensional process. Both graphs show the ratio of posterior standard deviation to the prior steady-state standard deviation of the process, along with the value of $c$ minimizing the posterior variance (blue line), and minimizing the reduction of posterior variance when $\mu_t = 0$ (yellow line). The process is initialized from its steady-state distribution. The posterior variance is estimated by averaging over the time interval $[5, 10]$ and across 1000 trials for each data point. Parameters for both graphs: $a = -1, d = 0.5, \sigma_{\mathrm{pop}}^2 = 0.1$. In the graph on the left, $\sigma_{\mathrm{tc}}^2 = 0.01$; on the right, $\lambda^0 = 50$.

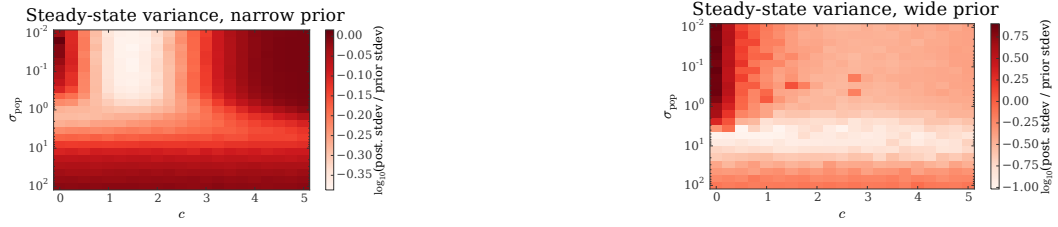

Figure 4: Optimal population distribution depends on prior variance relative to tuning curve width. A static scalar state drawn from $\mathcal{N}(0, \sigma_{\mathrm{p}}^2)$ is filtered with tuning curve $\sigma_{\mathrm{tc}} = 1$ and preferred stimulus density $\mathcal{N}(c, \sigma_{\mathrm{pop}}^2)$. Both graphs show the posterior standard deviation relative to the prior standard deviation $\sigma_{\mathrm{p}}$. In the left graph, the prior distribution is narrow, $\sigma_{\mathrm{p}}^2 = 0.1$, whereas on the right, it is wide, $\sigma_{\mathrm{p}}^2 = 10$. In both cases the filter is initialized with the correct prior, and the square error is averaged over the time interval $[5, 10]$ and across 100 trials for each data point.

Figure 4 (right), the prior is wide relative to the tuning curve width, leading to an optimal population with variance that roughly matches the prior variance. When both the tuning curves and the population density are narrow relative to the prior, so that spikes are rare (low values of $\sigma_{\mathrm{pop}}$ in Figure 4 (right)), the ADF approximation becomes poor, resulting in MSEs larger than the prior variance.

## 5  Conclusions

We have introduced an analytically tractable Bayesian approximation to point process filtering, allowing us to gain insight into the generally intractable infinite-dimensional filtering problem. The approach enables the derivation of near-optimal encoding schemes going beyond previously studied uniform coding assumptions. The framework is presented in continuous time, circumventing temporal discretization errors and numerical imprecisions in sampling-based methods, applies to fully dynamic setups, and directly estimates the MSE rather than lower bounds to it. It successfully explains observed experimental results, and opens the door to many future predictions. Future work will include a development of previously successful mean field approaches [13] within our more general framework, leading to further analytic insight. Moreover, the proposed strategy may lead to practically useful decoding of spike trains.

## Footnotes

[1]The model considered in [22] assumes linear dynamics and uniform coding – meaning that the total rate of $N_t$, namely $\int_\theta \lambda_t\left(\theta, X_t\right) d\theta$, is independent of $X_t$. However, these assumption are only relevant to establish other proposition in that paper. The proof of equation (3) still holds as is in our more general setting.

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
