[Supplementary Material]

# A Tractable Approximation to Optimal Point Process Filtering: Application to Neural Encoding – Supplementary Material

**Yuval Harel, Ron Meir**
Department of Electrical Engineering
Technion – Israel Institute of Technology
Technion City, Haifa, Israel
{yharel@tx,rmeir@ee}.technion.ac.il

**Manfred Opper**
Department of Artificial Intelligence
Technical University Berlin
Berlin 10587, Germany
opperm@cs.tu-berlin.de

## 1 Derivation of the ADF filtering equations for linear dynamics

### 1.1 Setting and notation

In the main text, we have presented our model in an open-loop setting, where the process $X$ is passively observed. Here we consider a more general setting, incorporating a control process $U_t$, so the dynamics are

$$dX_t = (A(X_t) + B(U_t)) \, dt + D(X_t) \, dW_t, \tag{1}$$

where, in general, $U_t$ is a function of $\mathcal{N}_t$.

For the purposes of the derivation, it is convenient to work with precision matrices rather than variance matrices. We write $F = \Sigma_{\text{pop}}^{-1}$, $R = \Sigma_{\text{tc}}^{-1}$ and $Q_t = \Sigma_t^{-1}$. Thus the density of the process $N$ at $(t, \theta)$ given $X_{[0,t]}, \mathcal{N}_t$ is

$$\lambda_t(\theta, X_t) = \lambda^0 \sqrt{\frac{|F|}{(2\pi)^m}} \exp\left( -\frac{1}{2} \|\theta - c\|_F^2 - \frac{1}{2} \|HX_t - \theta\|_R^2 \right), \tag{2}$$

$$= \lambda^0 \sqrt{\frac{|F|}{(2\pi)^m}} \exp\left( -\frac{1}{2} \|HX_t - c\|_M^2 - \frac{1}{2} \left\|\theta - (F+R)^{-1}(Fc + RHX_t)\right\|_{F+R}^2 \right), \tag{3}$$

where $M \triangleq \left(F^{-1} + R^{-1}\right)^{-1}$.

We denote by $P(\cdot, t)$ the posterior density of $X_t$ given $\mathcal{N}_t$, and by $\mathrm{E}_P^t[\cdot]$ the posterior expectation based on observations up to time $t$. We will simply write $\mathrm{E}_P[\cdot]$ when the time $t$ is obvious from context.

### 1.2 Filtering equations between spikes

#### 1.2.1 Exact filtering equations for the first two moments

As seen in [1], the PDE for the posterior density,

$$d_t P(x,t) = \mathcal{L}_t^* P(x,t) \, dt + P(x,t) \int_{\mathbb{R}^n} \frac{\lambda_t(\theta, x) - \hat{\lambda}_t(\theta)}{\hat{\lambda}_t(\theta)} \left( N(dt \times d\theta) - \hat{\lambda}_t(\theta) \, d\theta \, dt \right), \tag{4}$$

still holds in the closed-loop case. Here $\mathcal{L}_t$ is the posterior infinitesimal generator, defined with an additional conditioning on $\mathcal{N}_t$,

$$\mathcal{L}_t f(x) = \lim_{\Delta t \to 0^+} \frac{\mathrm{E}[f(X_{t+\Delta t}) | X_t = x, \mathcal{N}_t] - f(x)}{\Delta t},$$

and $\mathcal{L}_t^*$ is its adjoint. Note that in this closed-loop setting, the infinitesimal generator is itself a random operator, due to its dependence on past observations through the control law, and that $N_t$ is no longer a doubly-stochastic Poisson process.

*Between spikes*, (4) simplifies to

$$\frac{\partial}{\partial t} P(x,t) = \mathcal{L}_t^* P(x,t) - P(x,t) \int_{\mathbb{R}^n} \left( \lambda_t(\theta, x) - \hat{\lambda}_t(\theta) \right) d\theta,$$

so for a sufficiently well-behaved function $f$,

$$
\begin{aligned}
\frac{\partial \mathrm{E}_P\left[ f(X_t) \right]}{\partial t} &= \int f(x) \left( \mathcal{L}_t^* P(x,t) + P(x,t) \int \left( \hat{\lambda}_t(\theta) - \lambda_t(\theta, x) \right) d\theta \right) dx \\
&= \int P(x,t) \left( \mathcal{L}_t f(x) + f(x) \int \left( \hat{\lambda}_t(\theta) - \lambda_t(\theta, x) \right) d\theta \right) dx \\
&= \mathrm{E}_P\left[ \mathcal{L}_t f(X_t) + f(X_t) \int \left( \hat{\lambda}_t(\theta) - \lambda_t(\theta, X_t) \right) d\theta \right].
\end{aligned}
$$

Assuming the state evolves as in (1), the (closed loop) infinitesimal generator is

$$\mathcal{L}_t f(x) = (A(x) + B(U_t))^T \nabla f(x) + \frac{1}{2} \mathrm{Tr}\left[ \nabla^2 f(x) D(x) D(x)^T \right],$$

so, letting $\mu_t = \mathrm{E}_P X_t$, $\tilde{X}_t = X_t - \mu_t$, $\Sigma_t = \mathrm{E}_P\left[ \tilde{X}_t \tilde{X}_t^T \right]$,

$$
\begin{aligned}
\frac{d\mu_t}{dt} &= \mathrm{E}_P\left[ A(X_t) \right] + B(U_t) + \mathrm{E}_P\left[ X_t \int \left( \hat{\lambda}_t(\theta) - \lambda_t(\theta, X_t) \right) d\theta \right], \\
\frac{d\Sigma_t}{dt} &= \mathrm{E}_P\left[ A(X_t) \tilde{X}_t^T \right] + \mathrm{E}_P\left[ \tilde{X}_t A(X_t)^T \right] + \mathrm{E}_P\left[ D(X_t) D(X_t)^T \right] \\
&\quad + \mathrm{E}_P\left[ \tilde{X}_t \tilde{X}_t^T \int \left( \hat{\lambda}_t(\theta) - \lambda_t(\theta, X_t) \right) d\theta \right]. \quad (5)
\end{aligned}
$$

### 1.2.2 ADF approximation

The computations that follow frequently require multiplying Gaussian functions, sometimes with a possibly degenerate precision matrix. To this end, we use the following slightly generalized form of a well-known result about the sum of quadratic forms.

*Claim.* Let $x, a, b \in \mathbb{R}^n$ and $A, B \in \mathbb{R}^{n \times n}$ be symmetric matrices such that $A + B$ is non-singular. Then

$$\|x - a\|_A^2 + \|x - b\|_B^2 = \|a - b\|_{A(A+B)^{-1}B}^2 + \left\| x - (A+B)^{-1}(Aa + Bb) \right\|_{A+B}^2.$$

*Proof.* This is proved by a straightforward completion of squares. If $A, B$ are invertible,

$$
\begin{aligned}
\|x - a\|_A^2 + \|x - b\|_B^2 &= \|x\|_A^2 - x^T Aa - a^T Ax + \|a\|_A^2 + \|x\|_B^2 - x^T Bb - b^T Bx + \|b\|_B^2 \\
&= \|x\|_{A+B}^2 - x^T (Aa + Bb) - (Aa + Bb)^T x + \|a\|_A^2 + \|b\|_B^2 \\
&= \left\| x - (A+B)^{-1}(Aa + Bb) \right\|_{A+B}^2 - \left\| (A+B)^{-1}(Aa + Bb) \right\|_{A+B}^2 \\
&\quad + \|a\|_A^2 + \|b\|_B^2 \\
&= \left\| x - (A+B)^{-1}(Aa + Bb) \right\|_{A+B}^2 \\
&\quad + \underbrace{\|a\|_A^2 + \|b\|_B^2 - \|Aa + Bb\|_{(A+B)^{-1}}^2}_{*}
\end{aligned}
$$

$$
\begin{aligned}
* &= \|a\|_A^2 + \|b\|_B^2 - \|Aa\|_{(A+B)^{-1}}^2 - a^T A (A+B)^{-1} Bb - b^T B (A+B)^{-1} Aa - \|Bb\|_{(A+B)^{-1}}^2 \\
&= a^T A \left( a - (A+B)^{-1} Aa - (A+B)^{-1} Bb \right) + b^T B \left( b - (A+B)^{-1} Bb - (A+B)^{-1} Aa \right) \\
&= a^T A (A+B)^{-1} B (a-b) + b^T B (A+B)^{-1} A (b-a) \\
&= a^T \left( B^{-1} + A^{-1} \right)^{-1} (a-b) + b^T \left( A^{-1} + B^{-1} \right)^{-1} (b-a) \\
&= \|a-b\|_{(A^{-1}+B^{-1})^{-1}}^2 \\
&= \|a-b\|_{A(A+B)^{-1}B}^2
\end{aligned}
$$

By continuity, the claim also holds when $A, B$ are not both invertible, provided $(A+B)$ is invertible. $\qquad\square$

Computing the expectations in (5) involves computation of integrals containing $P(x,t)\lambda_t(\theta,x)$. Taking the ADF approximation $P(x,t) \approx \mathcal{N}(x; \mu_t, \Sigma_t)$, and using the claim above, we have

$$
\begin{aligned}
P(x,t)\lambda_t(\theta,x) &\approx \lambda^0 \sqrt{\frac{|F|}{(2\pi)^m}} \mathcal{N}(x; \mu_t, \Sigma_t) \\
&\quad \times \exp\left( -\frac{1}{2} \left( \|Hx - c\|_M^2 + \left\| \theta - (F+R)^{-1}(Fc + RHx) \right\|_{F+R}^2 \right) \right) \\
&= \lambda^0 \sqrt{\frac{|F|}{(2\pi)^{m+n}|\Sigma_t|}} \exp\left( -\frac{1}{2} \|x - \mu_t\|_{Q_t}^2 \right) \\
&\quad \times \exp\left( -\frac{1}{2} \left( \|x - \bar{c}\|_{H^T MH}^2 + \left\| \theta - (F+R)^{-1}(Fc + RHx) \right\|_{F+R}^2 \right) \right) \\
&= \lambda^0 \sqrt{\frac{|F|}{(2\pi)^{m+n}|\Sigma_t|}} \exp\left( -\frac{1}{2} \left( \|\mu_t - \bar{c}\|_{Q_t^M}^2 + \|x - \mu_t^M\|_{Q_t + H^T MH}^2 \right) \right) \\
&\quad \times \exp\left( -\frac{1}{2} \left\| \theta - (F+R)^{-1}(Fc + RHx) \right\|_{F+R}^2 \right)
\end{aligned}
$$

where $H_r^{-1}$ is any right inverse of $H$, and

$$
\begin{aligned}
\bar{c} &\triangleq H_r^{-1} c \\
M &\triangleq F(F+R)^{-1} R = \left( F^{-1} + R^{-1} \right)^{-1} \\
\mu_t^M &\triangleq \left( Q_t + H^T MH \right)^{-1} \left( Q_t \mu_t + H^T MH\bar{c} \right) \\
Q_t^M &\triangleq Q_t \left( Q_t + H^T MH \right)^{-1} H^T MH = \left( I + H^T MH\Sigma_t \right)^{-1} H^T MH.
\end{aligned}
$$

An alternate form for $Q_t^M$ may be derived from the Woodbury identity as follows,

$$
\begin{aligned}
Q_t^M &= \left( I + H^T MH\Sigma_t \right)^{-1} H^T MH \\
&= \left( I - H^T \left( M^{-1} + H\Sigma_t H^T \right)^{-1} H\Sigma_t \right) H^T MH \\
&= H^T \left( I - \left( M^{-1} + H\Sigma_t H^T \right)^{-1} H\Sigma_t H^T \right) MH \\
&= H^T \left( \left( M^{-1} + H\Sigma_t H^T \right)^{-1} M^{-1} \right) MH \\
&= H^T S_t^M H, \tag{6}
\end{aligned}
$$

where

$$
S_t^M \triangleq \left( M^{-1} + H\Sigma_t H^T \right)^{-1},
$$

so we can write

$$
\begin{aligned}
P(x,t)\lambda_t(\theta,x) &\approx \lambda^0 \sqrt{\frac{|F|}{(2\pi)^{m+n}|\Sigma_t|}} \exp\left( -\frac{1}{2} \left( \|H\mu_t - c\|_{S_t^M}^2 + \|x - \mu_t^M\|_{Q_t + H^T MH}^2 \right) \right) \\
&\quad \times \exp\left( -\frac{1}{2} \left\| \theta - (F+R)^{-1}(Fc + RHx) \right\|_{F+R}^2 \right).
\end{aligned}
$$

Now we define
$$g_t \triangleq \int \hat{\lambda}_t(\theta)\, d\theta = \int \mathrm{E}_P\left[\lambda_t(\theta, X_t)\right] d\theta,$$
and compute its value as follows:

$$
\begin{aligned}
g_t &= \int \hat{\lambda}_t(\theta)\, d\theta \\
&= \int\int P(x,t)\,\lambda_t(\theta, x)\, dx\, d\theta \\
&\approx \lambda^0 \sqrt{\frac{\det F}{(2\pi)^{m+n}\det \Sigma_t}} \exp\left(-\frac{1}{2}\|H\mu_t - c\|^2_{S_t^M}\right) \int dx \exp\left(-\frac{1}{2}\left\|x - \mu_t^M\right\|^2_{Q_t + H^T M H}\right) \\
&\quad\times \int d\theta \exp\left(-\frac{1}{2}\left\|\theta - (F+R)^{-1}(Fc + RHx)\right\|^2_{F+R}\right) \\
&= \lambda^0 \sqrt{\frac{\det F}{(2\pi)^n \det \Sigma_t \det(F+R)}} \exp\left(-\frac{1}{2}\|H\mu_t - c\|^2_{S_t^M}\right) \\
&\quad \int dx \exp\left(-\frac{1}{2}\left\|x - \mu_t^M\right\|^2_{Q_t + H^T M H}\right) \\
&= \lambda^0 \sqrt{\frac{\det F}{\det \Sigma_t \det(Q_t + H^T M H)\det(F+R)}} \exp\left(-\frac{1}{2}\|H\mu_t - c\|^2_{S_t^M}\right)
\end{aligned}
$$

To simplify the expression under the square root, we note that

$$\frac{\det F}{\det(F+R)} = \det\left(F(F+R)^{-1}\right) = \det\left(\left(R^{-1} + F^{-1}\right)^{-1} R^{-1}\right) = \frac{\det M}{\det R},$$

and, using the matrix determinant lemma and (6)

$$\det\left(Q_t + H^T M H\right) = \det Q_t \det M \det\left(M^{-1} + H\Sigma_t H^T\right) = \frac{\det M}{\det \Sigma_t \det S_t^M}$$

so

$$g_t = \lambda^0 \sqrt{\frac{\det S_t^M}{\det R}} \exp\left(-\frac{1}{2}\|H\mu_t - c\|^2_{S_t^M}\right).$$

A similar computation yields additional terms from (5), expressed in terms of $g_t$.

$$
\begin{aligned}
\mathrm{E}_P\left[X_t \int \lambda_t(\theta, X_t)\, d\theta\right] &= \int dx \int d\theta\, x\, P(x,t)\,\lambda_t(\theta, x) \\
&\approx \lambda^0 \sqrt{\frac{\det F}{(2\pi)^{m+n}\det \Sigma_t}} \exp\left(-\frac{1}{2}\|H\mu_t - c\|^2_{S_t^M}\right) \\
&\quad\times \int dx\, x \exp\left(-\frac{1}{2}\left\|x - \mu_t^M\right\|^2_{Q_t + H^T M H}\right) \\
&\quad\times \int d\theta \exp\left(-\frac{1}{2}\left\|\theta - (F+R)^{-1}(Fc + RHx)\right\|^2_{F+R}\right) \\
&= \lambda^0 \sqrt{\frac{\det F}{(2\pi)^n \det \Sigma_t \det(F+R)}} \exp\left(-\frac{1}{2}\|H\mu_t - c\|^2_{S_t^M}\right) \\
&\quad\times \int dx\, x \exp\left(-\frac{1}{2}\left\|x - \mu_t^M\right\|^2_{Q_t + H^T M H}\right) \\
&= \lambda^0 \sqrt{\frac{\det F}{\det \Sigma_t \det(Q_t + H^T M H)\det(F+R)}} \exp\left(-\frac{1}{2}\|H\mu_t - c\|^2_{S_t^M}\right)\mu_t^M \\
&= g_t \mu_t^M,
\end{aligned}
$$

$$\mathrm{E}_P\left[\tilde{X}_t \tilde{X}_t^T \int \lambda_t\left(\theta, X_t\right) d\theta\right] = \int dx \int d\theta\, P\left(x, t\right)\left(x-\mu_t\right)\left(x-\mu_t\right)^T \lambda_t\left(\theta, x\right) d\theta$$

$$\approx \lambda^0 \sqrt{\frac{\det F}{(2\pi)^{m+n} \det \Sigma_t}} \exp\left(-\frac{1}{2}\left\|H\mu_t - c\right\|_{S_t^M}^2\right)$$

$$\times \int dx\,\left(x-\mu_t\right)\left(x-\mu_t\right)^T \exp\left(-\frac{1}{2}\left\|x-\mu_t^M\right\|_{Q_t+H^T M H}^2\right)$$

$$\times \int d\theta \exp\left(-\frac{1}{2}\left\|\theta-(F+R)^{-1}\left(Fc+RHx\right)\right\|_{F+R}^2\right)$$

$$= \lambda^0 \sqrt{\frac{\det F}{(2\pi)^{n} \det \Sigma_t \det\left(F+R\right)}} \exp\left(-\frac{1}{2}\left\|H\mu_t - c\right\|_{S_t^M}^2\right)$$

$$\times \int dx\,\left(x-\mu_t\right)\left(x-\mu_t\right)^T \exp\left(-\frac{1}{2}\left\|x-\mu_t^M\right\|_{Q_t+H^T M H}^2\right)$$

$$= \lambda^0 \sqrt{\frac{\det F}{\det \Sigma_t \det\left(Q_t+H^T M H\right) \det\left(F+R\right)}}$$

$$\times \exp\left(-\frac{1}{2}\left\|H\mu_t - c\right\|_{S_t^M}^2\right)\left[\left(Q_t+H^T M H\right)^{-1}+\left(\mu_t-\mu_t^M\right)\left(\mu_t-\mu_t^M\right)^T\right]$$

$$= g_t\left[\left(Q_t+H^T M H\right)^{-1}+\left(\mu_t-\mu_t^M\right)\left(\mu_t-\mu_t^M\right)^T\right].$$

Assuming $X$ has linear dynamics, substituting these results into (5) yields the following filtering equations between spikes (we abuse notation and use $\mu_t, \Sigma_t$ to refer to the ADF-approximate quantities from here on),

$$\frac{d\mu_t}{dt} = A\mu_t + B\left(U_t\right) + g_t\left(\mu_t - \mu_t^M\right),$$

$$\frac{d\Sigma_t}{dt} = A\Sigma_t + \Sigma_t A^T + DD^T$$

$$+ g_t\left[\Sigma_t - \left(Q_t+H^T M H\right)^{-1} - \left(\mu_t-\mu_t^M\right)\left(\mu_t-\mu_t^M\right)^T\right]. \tag{7}$$

We simplify this by computing

$$\mu_t - \mu_t^M = \mu_t - \left(Q_t+H^T M H\right)^{-1}\left(Q_t\mu_t + H^T M H\bar{c}\right)$$

$$= \left(Q_t+H^T M H\right)^{-1} H^T M H\left(\mu_t-\bar{c}\right)$$

$$= \Sigma_t Q_t^M\left(\mu_t-\bar{c}\right)$$

$$= \Sigma_t H^T S_t^M\left(H\mu_t - c\right),$$

$$\Sigma_t - \left(Q_t+H^T M H\right)^{-1} = \Sigma_t\left(I-\left(I+H^T M H\Sigma_t\right)^{-1}\right)$$

$$= \Sigma_t H^T\left(M^{-1}+H\Sigma_t H^T\right)^{-1} H\Sigma_t$$

$$= \Sigma_t H^T S_t^M H\Sigma_t,$$

where we have used the Woodbury identity and (6). Substituting into (7) we obtain the form

$$\frac{d\mu_t}{dt} = A\mu_t + B\left(U_t\right) + g_t\Sigma_t H^T S_t^M\left(H\mu_t - c\right)$$

$$\frac{d\Sigma_t}{dt} = A\Sigma_t + \Sigma_t A^T + DD^T$$

$$+ g_t\Sigma_t H^T\left[S_t^M - S_t^M\left(H\mu_t - c\right)\left(H\mu_t - c\right)^T S_t^M\right] H\Sigma_t. \tag{8}$$

## 1.3 Effect of spikes

When a spike occurs at time $t$ in preferred location $\theta$, the update according to (4) is

$$
\begin{aligned}
P\left(x, t^{+}\right) &= P\left(x, t^{-}\right) + P\left(x, t^{-}\right) \frac{\lambda_{t-}(\theta, x) - \hat{\lambda}_{t-}(\theta)}{\hat{\lambda}_{t-}(\theta)} \\
&= P\left(x, t^{-}\right) \frac{\lambda_{t-}(\theta, x)}{\hat{\lambda}_{t-}(\theta)} \\
&= \frac{P\left(x, t^{-}\right) \lambda_{t-}(\theta, x)}{\int P\left(x, t^{-}\right) \lambda_{t-}(\theta, x)\, dx}.
\end{aligned}
$$

To compute this ratio we note that $P(x, t)\, \lambda_t(\theta, x)$, under the ADF approximation, may be written as a single Gaussian in $x$,

$$
\begin{aligned}
P(x, t)\, \lambda_t(\theta, x)\, dx &\approx \lambda^0 \sqrt{\frac{\det F}{(2\pi)^{m+n} \det \Sigma_t}} \exp\left(-\frac{1}{2}\|x - \mu_t\|_{Q_t}^2 - \frac{1}{2}\|\theta - c\|_F^2 - \frac{1}{2}\|Hx - \theta\|_R^2\right) \\
&= \lambda^0 \sqrt{\frac{\det F}{(2\pi)^{m+n} \det \Sigma_t}} \exp\left(-\frac{1}{2}\|\theta - c\|_F^2 - \frac{1}{2}\|x - \mu_t\|_{Q_t}^2 - \frac{1}{2}\|x - H_r^{-1}\theta\|_{H^T R H}^2\right) \\
&= C_t(\theta) \cdot \exp\left(-\frac{1}{2}\left\|x - \left(Q_t + H^T R H\right)^{-1}\left(Q_t \mu_t + H^T R \theta\right)\right\|_{Q_t + H^T R H}^2\right),
\end{aligned}
$$

where

$$
\begin{aligned}
C_t(\theta) &= \lambda^0 \sqrt{\frac{\det F}{(2\pi)^{m+n} \det \Sigma_t}} \exp\left(-\frac{1}{2}\|\theta - c\|_F^2 - \frac{1}{2}\|H_r^{-1}\theta - \mu_t\|_{Q_t^R}^2\right) \\
Q_t^R &\triangleq Q_t\left(Q_t + H^T R H\right)^{-1} H^T R H = \left(I + H^T R H \Sigma_t\right)^{-1} H^T R H.
\end{aligned}
$$

Analogously to the computation for $Q_t^M$ above, we have $Q_t^R = H^T S_t^R H$, where

$$
S_t^R \triangleq \left(R^{-1} + H \Sigma_t H^T\right)^{-1},
$$

Now $P\left(x, t^{+}\right)$ is given by the normalized Gaussian,

$$
\begin{aligned}
P\left(x, t^{+}\right) &= \frac{P\left(x, t^{-}\right) \lambda_{t-}(\theta, x)}{\int P\left(x, t^{-}\right) \lambda_{t-}(\theta, x)\, dx} \\
&= \sqrt{\frac{\det\left(\Sigma_{t-}^{-1} + H^T R H\right)}{(2\pi)^n}} \exp\left(-\frac{1}{2}\left\|x - \left(\Sigma_{t-} + H^T R H\right)^{-1}\left(\Sigma_{t-}^{-1}\mu_{t-} + H^T R \theta\right)\right\|_{\Sigma_{t-}^{-1} + H^T R H}^2\right) \\
&= \mathcal{N}\left(x, \left(\Sigma_{t-}^{-1} + H^T R H\right)^{-1}\left(\Sigma_{t-}^{-1}\mu_{t-} + H^T R \theta\right), \left(\Sigma_{t-}^{-1} + H^T R H\right)^{-1}\right),
\end{aligned}
$$

and the update is

$$
\begin{aligned}
\mu_{t+} &= \left(\Sigma_{t-}^{-1} + H^T R H\right)^{-1}\left(\Sigma_{t-}^{-1}\mu_{t-} + H^T R \theta\right) \\
\Sigma_{t+} &= \left(\Sigma_{t-}^{-1} + H^T R H\right)^{-1}.
\end{aligned}
$$

To incorporate these updates into the inter-spike SDE (8) they can be cast in the form

$$
\begin{aligned}
\mu_{t+} &= \mu_{t-} + \left(\Sigma_{t-}^{-1} + H^T R H\right)^{-1} H^T R H \left(H_r^{-1}\theta - \mu_{t-}\right) \\
&= \mu_{t-} + \Sigma_{t-} Q_{t-}^R \left(H_r^{-1}\theta - \mu_{t-}\right) \\
&= \mu_{t-} + \Sigma_{t-} H^T S_{t-}^R \left(\theta - H\mu_{t-}\right), \\
\Sigma_{t+} &= \Sigma_{t-} - \left(\Sigma_{t-}^{-1} + H^T R H\right)^{-1} H^T R H \Sigma_{t-} \\
&= \Sigma_{t-} - \Sigma_{t-} Q_{t-}^R \Sigma_{t-} \\
&= \Sigma_{t-} - \Sigma_{t-} H^T S_{t-}^R H \Sigma_{t-},
\end{aligned}
$$

giving the full filtering SDE

$$
\begin{aligned}
d\mu_t &= A\mu_t dt + B\left(U_t\right)dt + g_t \Sigma_t H^T S_t^M \left(H\mu_t - c\right)dt + \Sigma_{t-} H^T S_{t-}^R \int_{\theta \in \mathbb{R}^m} \left(\theta - H\mu_{t-}\right) N\left(dt \times d\theta\right), \\
d\Sigma_t &= \left(A\Sigma_t + \Sigma_t A^T + DD^T + g_t \Sigma_t H^T \left[S_t^M - S_t^M \left(H\mu_t - c\right)\left(H\mu_t - c\right)^T S_t^M\right] H\Sigma_t\right)dt \\
&\quad - \Sigma_{t-} H^T S_{t-}^R H\Sigma_{t-} dN_t.
\end{aligned}
\tag{9}
$$

## 2  Non-linear dynamics

In case of non-linear dynamics

$$
dX_t = \left(A\left(X_t\right) + B\left(U_t\right)\right)dt + D_t dW_t
$$

the ADF approximation may also be applied to the terms involving $A\left(X_t\right)$ in (5). Assume $A^{(i)}$, the $i$-th element of $A$, is given by a power series around $\mu_t$, written in multi-index notation,

$$
A^{(i)}\left(x\right) = \sum_\alpha A_\alpha^{(i)}\left(\mu_t\right)\left(x - \mu_t\right)^\alpha,
$$

where the sum is over all multi-indices $\alpha$. Then, assuming the ADF approximation $X_t \sim \mathcal{N}\left(\mu_t, \Sigma_t\right)$,

$$
\mathrm{E}_P\left[A^{(i)}\left(X_t\right)\right] = \sum_\alpha A_\alpha^{(i)}\left(\mu_t\right)\mathrm{E}_\alpha\left(\Sigma_t\right),
$$

where $\mathrm{E}_\alpha\left(\Sigma\right)$ is defined as $\mathrm{E}\left(Z^\alpha\right) = \mathrm{E}\prod_k Z_k^{\alpha_k}$ for $Z \sim \mathcal{N}\left(0, \Sigma\right)$, and may be computed from Isserlis' theorem. Similarly,

$$
\begin{aligned}
\mathrm{E}_P\left[A\left(X_t\right)\tilde{X}_t^T\right]_{ij} &= \mathrm{E}_P\left[A^{(i)}\left(X_t\right)\left(X_t^{(j)} - \mu_t^{(j)}\right)\right] \\
&= \sum_\alpha A_\alpha^{(i)}\left(\mu_t\right)\mathrm{E}_{\alpha + e_j}\left(\Sigma_t\right),
\end{aligned}
$$

where $e_j$ is $j$-th standard basis vector (the multi-index corresponding to the single index $j$).

Writing $A_\alpha = \left(A_\alpha^{(1)}, \ldots A_\alpha^{(n)}\right)^T$ and $\mathbf{E}_{\alpha,t} = \left(\mathrm{E}_{\alpha + e_1}\left(\Sigma_t\right), \ldots, \mathrm{E}_{\alpha + e_n}\left(\Sigma_t\right)\right)$ the filtering equations become

$$
\begin{aligned}
d\mu_t &= \sum_\alpha A_\alpha\left(\mu_t\right)\mathrm{E}_\alpha\left(\Sigma_t\right) + B\left(U_t\right)dt \\
&\quad + g_t \Sigma_t H^T S_t^M \left(H\mu_t - c\right)dt + \Sigma_{t-} H^T S_{t-}^R \int_{\theta \in \mathbb{R}^m}\left(\theta - H\mu_{t-}\right)N\left(dt \times d\theta\right) \\
d\Sigma_t &= \left(\sum_\alpha \left(A_\alpha\left(\mu_t\right)\mathbf{E}_{\alpha,t}^T + \mathbf{E}_{\alpha,t} A_\alpha\left(\mu_t\right)^T\right) + DD^T \right. \\
&\quad \left. + g_t \Sigma_t H^T \left[S_t^M - S_t^M \left(H\mu_t - c\right)\left(H\mu_t - c\right)^T S_t^M\right]H\Sigma_t\right)dt \\
&\quad - \Sigma_{t-} H^T S_{t-}^R H\Sigma_{t-} dN_t.
\end{aligned}
$$

Analogous comments apply when the noise gain $D_t$ is a non-linear function $D\left(X_t\right)$, provided each element $\left[D\left(x\right)D\left(x\right)^T\right]_{ij}$ may be expanded as a power series.

## 3  Comparison of estimated posterior variance and MSE

In the main text, we studied optimal encoding using the posterior variance as a proxy for the MSE. Letting $\mu_t, \Sigma_t$ denote the approximate posterior moments given by the filter, the MSE and posterior variance are related as follows,

$$
\begin{aligned}
\mathrm{MSE}_t &\triangleq \mathrm{E}\left[\mathrm{tr}\left(X_t - \mu_t\right)\left(X_t - \mu_t\right)^T\right] = \mathrm{E}\mathrm{E}_P^t \mathrm{tr}\left(X_t - \mu_t\right)\left(X_t - \mu_t\right)^T \\
&= \mathrm{E}\left[\mathrm{tr}\left(\mathrm{Var}_P^t X_t\right)\right] + \mathrm{E}\left[\mathrm{tr}\left(\mu_t - \mathrm{E}_P^t X_t\right)\left(\mu_t - \mathrm{E}_P^t X_t\right)^T\right],
\end{aligned}
$$

Figure 1: Posterior variance vs. MSE when filtering a one-dimensional process $dX_t = -0.1X_t dt + 0.5 dW_t$ (the steady-state variance of this process is $\sigma_0^2 = 1.25$). The top plot shows the MSE and mean posterior variance. The bottom plot shows the ratio of means $\mathrm{MSE}/\langle\Sigma_t\rangle$ and the mean ratio $\langle\mathrm{SE}/\Sigma_t\rangle$ where SE is the squared error $(\mu_t - X_t)^2$. Sensory parameters are $c = 0, \sigma_{\mathrm{pop}}^2 = 0.1, \sigma_{\mathrm{tc}}^2 = 0.01, \lambda^0 = 10$. The means were taken across 1000 trials. Shaded areas indicate error estimates obtained as sample standard deviation divided by square root of number of trials.

(a)

(b)

Figure 2: Mean Square Error as a function of model parameters. This figure is based on the same data as Figure 3 in the main text, with Root Mean Square Error (RMSE) plotted instead of estimated posterior variance. See Figure 3 of main text for more details.

where $\mathrm{E}_P^t[\cdot], \mathrm{Var}_P^t[\cdot]$ are resp. the mean and covariance conditioned on $\mathcal{N}_t$, and $\mathrm{tr}$ is the trace operator. Thus for an exact filter, having $\mu_t = \mathrm{E}_P^t X_t, \Sigma_t = \mathrm{Var}_P^t X_t$, we would have $\mathrm{MSE}_t = \mathrm{trace}[\mathrm{E}(\Sigma_t)]$. Conversely, if we find that $\mathrm{MSE}_t \approx \mathrm{trace}[\mathrm{E}\Sigma_t]$, it suggests that the errors are small (though this is not guaranteed, since the errors in $\mu_t$ and $\Sigma_t$ may effect the MSE in opposite directions, if the variance is underestimated).

Figure 1 shows the variance and MSE in estimating a linear one-dimensional process, after averaging across 1000 trials. Although the posterior variance is, on average, overestimated at the start of trials, in the steady state it approximates the square error reasonably well.

We also show here variants of the Figures 3 and 4 from the main text (Figures 2 and 3, respectively), showing the MSE rather than the variance. The results look similar but noisier, except in Figure 3b for small population variance, where the ADF estimation is poor due to very few spikes occurring.

Figure 3: Optimal population distribution depends on prior variance relative to tuning curve width. This figure is based on the same data as Figure 4 in the main text, with MSE plotted instead of estimated posterior variance. See Figure 4 of main text for more details.

# References

[1] I. Rhodes and D. Snyder. Estimation and control performance for space-time point-process observations. *IEEE Transactions on Automatic Control*, 22(3):338–346, 1977.