[Reviews · NeurIPS 2015]

Submitted by Assigned_Reviewer_1

This paper describes a tractable method for performing dynamic state estimation from point process observations. The authors derive closed-form expressions for the mean and variance of a Gaussin approximation to the posterior (using assumed density filtering), characterize the optimal placement of tuning curves, and emphasizes the importance of information carried by "silences" to the posterior.

The paper is well-written, original, highly novel, and provides a very beautiful extension of previous work on optimal tuning curves for static problems. It is likely to have a substantial impact on the field and should certainly be accepted.

Comments:

1. The abstract and intro seem slightly misleading in implying that the solution is fully general, when the details presented here seem to require Gaussian tuning curves.

(Perhaps I missed it, but do the authors have a solution for dealing with non-Gaussian tuning curves, or a posterior density that is not Gaussian?

Even tuning curves given by a Gaussian plus a constant DC baseline rate would seem to create a big problem because one would no longer obtain a closed-form update to the posterior from the spike-related likelihood terms).

I still think the work is extremely nice, but a little bit more of a disclaimer about what cases are considered and/or limits on the form of the code might be helpful here.

(Alternatively, if they have an idea about how to handle non-zero baseline spike rate, it would helpful to hear it.)

2. I'm a little bit confused about one aspec of the Gaussian approximation to the non-spiking terms in the log-likelihood.

If we go a long time without seeing any spikes, the true posterior becomes bimodal due to the -f(\theta) term -- intuitively, the probability that the state is far away on one tail or the other becomes increasingly high, the longer we go without spikes.

What happens to the approximation in this case - does it simply tend towards uniform?

I assume this is an extreme case that the authors regard as relatively unlikely for a reasonable sized population with reasonable marginal spike rate, but would be nice to say a word or two about when the approximation will break down.

3.

Two refs the authors might consider citing:

- Huys, Zemel, Natarajan, & Dayan, NC 2007 - relatively early paper considering the problem of decoding a dynamic posterior distribution from spike trains.

- Grabska-Barwinska & Pillow, NIPS 2014 - considered optimal tuning curve problem for uniform case but for different error metrics & constraints.

Minor:

Fig 2 caption:

"dashed green line" - doesn't appear to be green.
Summary: Well-written, highly novel, and provides a very beautiful extension of previous work on optimal tuning curves. Very worthwhile both for the solution provided (i.e., how to tractably decode in such a setting) and for the theoretical treatment of optimal tuning curves for a more interesting problem than the classic "decoding a static stimulus" case treated in most of this literature.

Submitted by Assigned_Reviewer_2

Paper titled "An Analytically Tractable Bayesian Approximation to Optimal Point Process Filtering" presents a significant and novel results in optimal neural coding. It is an improvement of Snyder and coworker's old results on optimal point process filtering which had an assumption that the observed processes are uniformly distributed to represent any latent state with a constant firing rate over the population. We do not know if this is the case in the brain for sure, but it is highly plausible that certain latent states are not coded with constant population rates. Authors don't mention this, but because of energy constraints, it is useful to represent the most abundant states with lower firing rates.

Authors use assumed density filtering to find an approximate Gaussian process posterior (filtering only, not smoothing). This in turn gives us insight into non-uniform population rate coding. The results are very nice, well written, and could have high impact in the neural coding community.

This is written to be purely theoretical, and the method seems difficult to apply to actually use it as a point process filtering algorithm (or at least the paper is not focused on solving such practical problem). Hence, the title of the paper is a little bit misleading. This reviewer thinks the title can be updated to put more emphasis on the theoretical contribution to optimal neural coding better and less on the optimal filtering.

* p6. Figure (2) -> Figure 2
Summary: Well written paper with new exciting results on optimal neural coding.

Submitted by Assigned_Reviewer_3

The author(s) developed an analytically tractable Bayesian approximation to optimal filtering based on point process observations. The problem is originally intractable infirnite-dimensional continuous time filtering. To solve the evolution equation for the mean and covariance of the state, they suggested an approximated analytical method based on Assumed Density Filtering (ADF) with the bGaussian density. It turned out that the ADF approximation method is superior to Mean Square Error (MSE) optimal decoder. It was demonstrated that absence of spikes is informative. Construction of a theory seems solid, and the proposed method achieved the high performance.
Summary: The author(s) introduced an analytically tractable Bayesian approximation to point process filtering. Construction of a theory seems solid, and the proposed method achieved the high performance.

Author Feedback
Author rebuttal: Reviewer 1

Reviewer 1 wrote "The abstract and intro seem slightly misleading in implying that the solution is fully general, when the details presented here seem to require Gaussian tuning curves". Our analytic solution does indeed require Gaussian tuning curves, and we will modify the abstract and introduction to reflect this fact.

The reviewer also suggests that we mention cases where the ADF approximation breaks down. We did observe problems with the approximation in the scenario that the reviewer has mentioned - a long period without spikes, where the true posterior should become bimodal. We briefly mention this case in lines 403-405 and in the supplementary material on line 431 when explaining the poor performance of the filter for some parameter values as seen in Figure 4.

Reviewer 2

Reviewer 2 wrote that "the method seems difficult to apply to actually use it as a point process filtering algorithm", and suggested that the title be updated "to put more emphasis on the theoretical contribution to optimal neural coding", rather than optimal filtering. We agree that neural encoding should be mentioned in the title, and will modify the title accordingly. The article is indeed focused on theoretical rather than algorithmic issues.

However, we do not see any difficulty in applying the method as a filtering algorithm, given that the observation point process is well-described by the model. Indeed, we have used the method to filter point process observations in the various simulations described in the article and reflected in all figures.

Reviewer 4

Reviewer 4 has suggested that "the author would use their framework to track a signal in a standard Kalman-like filtering framework but based on point process observations". All the simulations described in the article are of this kind: they include a process with linear dynamics (in Figure 4, the degenerate case of a static "process"), filtered from point process observations.